# Return of the Neurotropic Enteroviruses: Co-Opting Cellular Pathways for Infection

**DOI:** 10.3390/v13020166

**Published:** 2021-01-22

**Authors:** Christine E. Peters, Jan E. Carette

**Affiliations:** Department of Microbiology and Immunology, Stanford University School of Medicine, Stanford, CA 94305, USA; c9peters@stanford.edu

**Keywords:** picornaviruses, enteroviruses, neuropathogenesis, virus–host interactions, EV-D68, EV-A71, CRISPR and haploid screens, host-directed therapeutics, antivirals, viral tissue tropism

## Abstract

Enteroviruses are among the most common human infectious agents. While infections are often mild, the severe neuropathogenesis associated with recent outbreaks of emerging non-polio enteroviruses, such as EV-A71 and EV-D68, highlights their continuing threat to public health. In recent years, our understanding of how non-polio enteroviruses co-opt cellular pathways has greatly increased, revealing intricate host–virus relationships. In this review, we focus on newly identified mechanisms by which enteroviruses hijack the cellular machinery to promote their replication and spread, and address their potential for the development of host-directed therapeutics. Specifically, we discuss newly identified cellular receptors and their contribution to neurotropism and spread, host factors required for viral entry and replication, and recent insights into lipid acquisition and replication organelle biogenesis. The comprehensive knowledge of common cellular pathways required by enteroviruses could expose vulnerabilities amenable for host-directed therapeutics against a broad spectrum of enteroviruses. Since this will likely include newly arising strains, it will better prepare us for future epidemics. Moreover, identifying host proteins specific to neurovirulent strains may allow us to better understand factors contributing to the neurotropism of these viruses.

## 1. Introduction

Poliovirus (PV), which once caused severe neuropathogenesis, is now on the verge of eradication following widespread vaccination. As a result, the majority of recent enterovirus (EV) infections associated with severe neuropathogenesis is due to outbreaks of emerging non-polio enteroviruses (NPEVs), highlighting their continuing threat to public health [1,2]. *Enterovirus*, a genus of the *Picornaviridae* family, is a diverse group of small non-enveloped viruses containing a single positive-strand (+) RNA genome. EVs are transmitted through respiratory or fecal–oral routes of infection, and can disseminate from their primary site of infection in the gastrointestinal (GI) or respiratory tract, spreading to infect other tissues and organs. EVs cause a wide range of diseases such as the common cold, bronchiolitis, pneumonia, myocarditis, and hand-foot-and-mouth disease (HFMD). Although NPEV infections are often mild, they are important neurovirulent pathogens and can cause severe central nervous system (CNS) diseases such as acute flaccid myelitis (AFM), a polio-like paralysis, and encephalitis [3]. Additionally, NPEVs are the main cause of aseptic meningitis [4]. EV-A71 and other enterovirus A pathogens were only recently recognized as a significant cause of severe CNS disease [2]. EV-A71 is a common cause of HFMD in infants and children, and can cause life-threatening brainstem encephalitis, meningitis, and polio-like paralysis. EV-A71 is endemic to the Asia-Pacific region, which has seen an increase in the number of infections in the last two decades with yearly large-scale outbreaks. EV-D68, another emerging NPEV, was considered to be a rare cause of respiratory illness before the early 2000s. However, in 2014, EV-D68 was identified as the causal agent of an outbreak of severe respiratory disease in children that was associated with an increase in cases of AFM. Since then, cases of EV-D68 respiratory disease and associated AFM have increased worldwide with outbreaks of EV-D68, following a biennial, seasonal pattern. In addition to EV-D68 and EV-A71, other NPEVs associated with neuropathogenesis include EV-D70, echoviruses, enterovirus C viruses such as CV-A24, coxsackie B viruses such as CV-B3, and coxsackie A viruses such as CV-A16 [3]. The increasing threat to human health posed by EVs underscores the importance of countermeasures needed to target this broad group of viruses. With the exception of vaccines against PV and two vaccines against EV-A71 recently brought to market in China, there are no approved antivirals or therapeutics to treat or prevent enterovirus infections. 

As obligate intracellular pathogens, viruses have evolved a variety of strategies to manipulate the host cellular machinery to promote their replication as well as to counteract host immune defenses. Due to their limited genome capacity, EVs depend on host cellular factors to accomplish distinct steps in their life cycle. EVs exploit cellular receptors for entry (Figure 1) [5], hijack host ribosomes and cellular proteins to translate and help replicate their genome, assemble new virions, and promote their spread. Each stage of the viral life cycle is highly dependent upon the interplay between viral proteins and host factors. Traditionally, antiviral drug discovery has focused on molecules targeting viral factors. Host-directed therapy (HDT) is an emerging strategy in the field of antivirals that aims to control viral infections by interfering with host cellular pathways [6]. Therapeutics that inhibit host cellular factors critical for infection by multiple viruses rather than viral proteins have the potential to be broader spectrum and may present a higher genetic barrier to the development of drug resistance. While resistance to drugs targeting viral factors can rapidly emerge due to the fast mutation rate of the viral genome, resistance to host-targeting drugs would require the virus to utilize an alternative host factor or become less dependent on the host factor for their infection. Although this is possible, these resistance mutations would likely not overlap with those that occur with direct antiviral therapeutics. Thus, the many cellular factors utilized by viruses can provide novel modes of action that can complement direct antivirals in combination therapies.

EVs are an attractive target for the development of HDT due to the presence of a large number of viral serotypes causing similar disease pathologies and the use of common cellular pathways to promote their infections. Development of HDT against EVs requires a comprehensive understanding of the host cellular factors that lead to viral replication and spread. Recent advancements in high-throughput methods have expanded our toolkit for studying host–virus interactions and paved the way for identification of host factors that promote and restrict EV infection. CRISPR-Cas9 and haploid genetic screens have been engineered to efficiently induce knockout (KO) mutations in almost any cell type, allowing for the study of host factors that are important for virus replication. Genome-scale genetic screens provide an unbiased and comprehensive way to identify pro-viral factors and have been successfully used to identify host factors required for many different viruses [7,8]. Advances in mass spectrometry-based proteomics have also allowed us to make detailed networks of host–viral protein–protein interactions and analyze the host and viral proteome in response to infection [9]. In this review, we focus on newly identified mechanisms by which EVs hijack the cellular machinery to promote their replication and spread and address their potential for the development of HDT. Understanding how this broad class of viruses depends on common cellular pathways uncovers basic aspects of cellular biology, reveals intricate host virus relationships, and leads to possible targets for host-directed antiviral therapy that will be critical to combat the rising threat NPEVs pose to human health.

## 2. The Critical Role of Cellular Receptors in Viral Entry

Entry of EVs into host cells is initiated by virion attachment to the cell surface through binding of host receptors, which leads to internalization of the virion by an endocytic mechanism [10]. Subsequently, uncoating cues such as low pH and engagement with an uncoating receptor induce conformational changes in the capsid that allow for the release of the viral RNA genome into the cytosol of the host cell [11]. PV, the most well studied enterovirus, utilizes one receptor protein (PVR) to mediate the binding of the virions to the cells, and to serve as uncoating receptor. Moreover, PVR’s role is non-redundant and cells lacking PVR are completely refractory to infection. This is not always the case; some viruses use distinct receptors to promote infection of different cell types, and some cellular receptors primarily mediate binding to the cell but do not initiate endocytosis or uncoating of the virion. One of the determining factors of susceptibility to viral infection is the availability of specific receptors on the surface of the host cell. Viral receptors can determine species-specific and tissue-specific tropism. Furthermore, they are attractive therapeutic targets, because they mediate the first necessary steps of virus infection, are rate-limiting, and can restrict infection in certain tissues. Receptor usage plays a critical role in dissemination of the virus throughout a host and disease pathogenesis. Despite the importance of viral receptors in the enterovirus life cycle, it is still poorly understood how different receptors contribute to viral infection of specific tissues. In recent years largely due to new high-throughput techniques, several new enterovirus receptors have been identified.

## 3. EV-D68 Receptors

As a respiratory pathogen, EV-D68 primarily infects the respiratory tract; however, EV-D68 has been increasingly linked to infection of the CNS leading to AFM [12,13]. Sialic acid and intercellular adhesion molecule 5 (ICAM-5) have been identified as cellular receptors for EV-D68. Sialic acids are monosaccharides terminally conjugated onto glycoproteins and glycolipids and are widely distributed throughout most tissues. Sialic acid molecules are highly abundant in the human respiratory tract and serve as receptors for many respiratory viruses including influenza. Many EV-D68 strains have been shown to bind to sialic acid [14,15], and treatment with neuraminidase, which cleaves terminal sialic acid, of various cell types decreases virion attachment and leads to a reduction in viral infection [14,15,16]. EV-D68 infection can be reduced by inhibition of sialic acid biosynthesis, conjugation of sialic acid onto glycans, or presentation of sialylated glycoproteins on the cell surface [14,17]. Furthermore, binding of sialic acid to EV-D68 Fermon has been shown to initiate a conformational change in the capsid proteins that facilitates destabilization of the capsid and uncoating of the virion [16]. EV-D68 does not bind to all sialylated glycoproteins, suggesting that the specific composition of the underlying sialylated glycan or protein is important to infection [14]. Interestingly, many contemporary clinical EV-D68 strains are not dependent on sialic acid for entry, as they have been shown to infect cell lines in the absence of sialic acid [14,15].

Several EVs use cell surface proteins belonging to the immunoglobulin superfamily as their receptor. This includes the major rhinovirus receptor, ICAM-1, the poliovirus receptor, PVR (CD155), and the coxsackie B virus receptor, CAR (Figure 1). Normally involved in diverse functions including cell–cell adhesion and immune responses, viruses have evolved to engage them to facilitate entry [18]. ICAM-5 was identified as a receptor for EV-D68 based on its similarity to the receptor of the related rhinoviruses (ICAM-1), and its expression pattern in cancer cell lines known to be susceptible or refractory for EV-D68 [19]. It was shown that the EV-D68 Fermon strain could bind to soluble ICAM-5, and knockdown of ICAM-5 decreased infection in certain human cell lines. Conversely, expression of ICAM-5 in non-permissive cells strongly enhanced infection by EV-D68 strains including contemporary strains. Soluble ICAM-5 reduced infection in cell culture when added to the medium and in vivo upon co-administration in an intracerebral (IC) neonatal mouse model of infection [19]. Based on these results and the observation that it is specifically expressed on the surface of telencephalic (the anterior portion of the forebrain) neurons of the CNS, it was proposed that ICAM-5 is a neuron-specific functional entry receptor for EV-D68 [18]. Incubation of EV-D68 with soluble ICAM-5 at pH 6.0 triggered particle uncoating, suggesting that it can act as an uncoating receptor, similar to sialic acid. Cryogenic electron microscopy (cryoEM) confirmed that soluble ICAM-5 mediates the transition from mature virions to uncoating intermediates [20]. Intriguingly, no structures could be resolved of the viral particle with ICAM-5, suggesting that the binding of ICAM-5 with the capsid might be transient and/or low affinity. 

## 4. EV-A71 Receptors

EV-A71, as an enteric virus, is transmitted by the fecal–oral route and infects a wide range of tissue types to cause HFMD, aseptic meningitis, and sometimes fatal brainstem encephalitis [3]. Scavenger receptor class B, member 2 (SCARB2), has been identified as the major uncoating receptor for EV-A71 infection [21]. SCARB2 is a ubiquitously expressed type III transmembrane protein belonging to the CD36 family of scavenger receptor proteins. SCARB2 is involved in the regulation of endosomal and lysosomal membrane transport and is primarily localized to the lysosomal membrane, with a small fraction localized to the plasma membrane. For EV-A71 and a number of other enterovirus A species including CV-A16, SCARB2 supports attachment and internalization, and induces conformational changes in the virion upon low pH [21,22]. It is the only receptor to date known to support binding, internalization, and uncoating of EV-A71. However, additional surface receptors have been described that support EV-A71 entry. It was proposed that these surface proteins are involved in a SCARB2 dependent entry mechanism by attaching, internalizing, and delivering EV-A71 to endosomal SCARB2 [23]. 

P-selectin glycoprotein 1 (PSGL-1) is another well studied receptor for EV-A71, which is expressed primarily on leukocytes. PSGL-1 regulates immune cell trafficking and plays a role in binding leukocytes to endothelial cells and platelets. While PSGL-1 binds to EV-A71 with high affinity, it does not induce conformational changes in the EV-A71 virion that leads to uncoating, acting primarily as an attachment factor [22]. 

In addition to SCARB2, Cyclophilin A (CypA) is the only other entry factor described to have uncoating activity. CypA was demonstrated to bind to the VP1 capsid protein and can initiate uncoating of EV-A71 virions in a pH-dependent manner [24]. In addition, knockdown of CypA or inhibition of CypA leads to reduced EV-A71 infection. It is suggested CypA functions as an uncoating regulator during EV-A71 internalization, and may aid other attachment factors in promoting effective cell entry.

In the absence of SCARB2, human tryptophanyl-tRNA synthetase (WARS) can make cells susceptible to infection by EV-A71 [25]. It was found that WARS is detected at the plasma membrane where it can directly interact with EV-A71 virions and facilitate entry. In addition to playing a role in EV-A71 entry, WARS was also found to be important for infection of other enterovirus species including EV-D68 and CV-A16. Interestingly, even in the presence of SCARB2, knockdown of WARS in a human rhabdomyosarcoma cell line (RD) or knockout of WARS in a human neuronal cell line (NT2) still inhibited effective EV-A71 infection. WARS was found to be an interferon (IFN) inducible entry factor for EV-A71 with interferon gamma (IFN-γ), a type II interferon, upregulating cell surface expression of WARS. EV-A71 probably facilitates cell surface expression of WARS through IFN signaling to enhance spread. The mechanism of WARS induced entry is unclear; it is still unknown if cell surface WARS can uncoat the virion independently or if it facilitates a SCARB2 independent entry route.

Recently, a role for cell surface prohibitin (PHB) was identified in EV-A71 entry, specifically into neuronal cells. Knockdown of PHB in mouse motor neuron cells (NSC-34) and a human neuroblastoma cell line (SK-N-SH) reduced EV-A71 infection, while knockdown of PHB in RD cells showed no effect on EV-A71 infection [26]. The capsid of EV-A71 was found to interact with cell surface PHB. Additionally, pretreatment with an antibody against PHB inhibited EV-A71 infection, suggesting that PHB acts at the entry stage of the life cycle [26]. Depletion of PHB also decreased viral replication in a replicon assay, suggesting its role may extend beyond promoting viral entry.

Several additional cell surface molecules, such as heparan sulfate (HS) [27], sialylated glycans, vimentin [28], nucleolin, annexin A2 (ANXA2), and fibronectin, have been shown to promote EV-A71 entry [23]. These molecules are not specific to EV-A71 infection and serve as receptors for a wide range of pathogens including bacteria, viruses, and parasites. It is likely that their main role in EV-A71 entry is to promote cell adsorption rather than acting as functional entry receptors that initiate uncoating.

## 5. Receptors for Other Neurotropic Enteroviruses

Enteroviruses are the leading cause of viral meningitis, and EVs as a whole are increasingly recognized as a significant cause of neuropathogenesis. Several essential receptors for NPEVs have been recently identified. Through an unbiased genome-wide genetic screen, KREMEN1, a transmembrane protein involved in WNT signaling, was identified as an entry receptor for multiple enterovirus A species including CV-A10 [29]. KREMEN1 is essential for infection of human cell lines, binds to CV-A10 virions at the cell surface, and promotes virus internalization. Furthermore, KREMEN1 deficiency protects mice from CV-A10 infection. Subsequent structural and biochemical studies showed that KREMEN1 is a functional uncoating receptor [30]. Incubation of CV-A10 with soluble KREMEN1 triggers the transition from mature virions to uncoating intermediates, a process that is accelerated by acidic conditions [30]. 

Until recently, the primary uncoating receptor for echoviruses, which belong to the enterovirus B species and are a major cause of aseptic meningitis, remained unknown. Decay accelerating factor (DAF) is a receptor for some echoviruses, since it has been shown to promote virion binding to the cell surface [31]. However, cryoEM studies demonstrated that DAF was unable to mediate particle uncoating [32], suggesting that DAF primarily functions to facilitate attachment. Additionally, DAF expression in nonpermissive cells did not sensitize the cells to infection [33], suggesting the usage of an additional receptor to promote entry. In multiple studies, human neonatal Fc receptor (FcRn) was identified as the uncoating receptor for a major group of echoviruses [34,35] (Figure 1). Unlike DAF, expression of FcRn in non-permissive cells rendered them susceptible to infection, and incubation of virions with FcRn induced conformational changes allowing for particle uncoating under acidic conditions [34].

## 6. Contribution of Viral Receptors to Neurotropism

As respiratory or fecal–oral pathogens, EVs first enter into and replicate at their primary sites of infection in the respiratory or gastrointestinal tract. Neuro-invasion involves dissemination of the virus from these initial sites of infection to other tissues and organs including the CNS [36]. There are multiple routes by which EVs could gain access into the CNS. One possibility is the virus enters by crossing the blood–brain barrier (BBB), either by penetrating the barrier directly, infecting brain endothelial cells, or through a Trojan horse approach by infecting immune cells that can cross the barrier. Another possibility is that the virus gains access to the CNS by infecting peripheral nerves and invades the CNS via retrograde axonal transport. It is also possible that viruses may use a combination of these approaches to access the CNS. In all cases, the virus must first enter into an extraneural compartment, such as the blood or skeletal muscle, in order to reach a site of entry into the CNS. Once the virus has penetrated the CNS, infection of particular regions of the CNS results in the clinical symptoms associated with neuropathies. One of the determining factors of tissue susceptibility to viral infection is the availability of specific receptors on the surface of the host cell. EVs may utilize multiple receptors and attachment factors to promote their infection of different tissues and aid in their spread throughout the host. Elucidating what tissues are targeted by EVs and what receptors they use to disseminate into various tissues is critical to understanding their disease pathogenesis and may offer potential therapeutic targets for preventing spread to the CNS. Host determinants of neurotropism remain ill-defined to date, and additional research on host factors that contribute to viral susceptibility and the clinical symptoms of disease will lead to important insights into the prevention of neurological complications.

Several notable observations into the mechanism of EV-D68 spread and neurotropism have been gained from mouse models. In recent studies, intracerebral (IC), intramuscular (IM), and intraperitoneal (IP) inoculations in neonatal mice have been able to support EV-D68 systemic infection and neuroinvasion leading to paralysis, with the highest levels of viral replication in the lung, muscle, and spinal cord [37]. Intranasal (IN) inoculation, which most closely mimics the natural route of infection, also led to systemic infection with rare paralysis consistent with the low rates of AFM in human populations [37]. Surprisingly, even after direct IC infection, no viral replication was observed in the brain, with infectious virus rapidly declining after initial inoculation. Moreover, EV-D68 infection was more efficient with IM injection than IC, leading to rapid viral replication in the muscle and spinal cord [36]. EV-D68 viral antigens in the CNS were found almost exclusively in motor neurons, suggesting that EV-D68 has a strong tropism for motor neurons over neurons in general [37]. Paralysis following EV-D68 infection is attributed to motor neuron injury and cell death due to viral replication in these cells. Retrograde transport from the muscle to motor neurons is likely the most efficient pathway to invade the CNS for EV-D68. While dissemination from the lungs to the muscle and subsequently the CNS may be the primary pathway by which EV-D68 accesses motor neurons, this does not rule out the possibility that EV-D68 can also access the CNS by crossing the BBB during a natural infection. 

The identification of sialic acid and ICAM-5 as functional receptors raises the question of whether these receptors contribute to neurotropism and organismal spread of EV-D68. To investigate this, Hixon and colleagues used human iPSC-derived motor neurons in a microfluidic chamber to model motor neurons in the spinal cord and their distal axons [38]. It was shown that contemporary EV-D68 strains were still able to undergo retrograde transport in the absence of sialic acid. Because ICAM-5 was not expressed on the distal axons of motor neurons, it was suggested that EV-D68 utilizes an entry mechanism independent of ICAM-5 and sialic acid to initiate retrograde transport [38]. While contemporary strains still retain the ability to bind to sialic acid [14], it is unclear if they can use sialic acid as a functional uncoating receptor to promote their dissemination and spread. The main evidence that sialic acid functions as an uncoating receptor has been of studies using the prototype EV-D68 strain (Fermon; isolated in 1962), which does not cause paralysis in an IC mouse model of infection [37]. It is possible that contemporary strains retain the ability to use sialic acid for entry into certain cell types, such as in the respiratory tract where sialic acid is highly abundant, but use additional receptors for CNS invasion and dissemination to different tissues. Since ICAM-5 is specifically expressed on the surface of telencephalic neurons of the CNS, it is possible that ICAM-5 plays a role in infection of neuronal compartments once the virus reaches the CNS, but is not likely to be involved in dissemination of EV-D68 to the CNS.

Other factors in addition to receptor usage may determine neurotropism of EV-D68. Since AFM has only been recently associated with EV-D68 infections, it is suspected that neurotropism is a newly acquired phenotype of recent strains. Both contemporary and prototypical strains have been shown to replicate in human cortical neurons, motor neurons, and astrocytes in tissue culture [15,38]. However, only contemporary 2014 outbreak strains have been shown to enter and replicate in the spinal cord causing paralysis in an IM mouse model of infection [39]. This result suggests that prototypical strains may not be able to disseminate and enter into the CNS during a natural route of infection. It is unclear if the lack of progression to the CNS of some strains in a mouse model of infection is due to differences in receptor usage, lower levels of replication in extraneural tissues inhibiting the ability to disseminate to the CNS, the usage of neuron specific host factors, or due to the ability to overcome host immune responses. Understanding the route of dissemination to the CNS and receptor or host determinants of EV-D68 neurotropism should be the focus of future research.

A combination of several receptors utilized by EV-A71 might contribute to EV-A71 neurotropism and pathogenesis. The current model of EV-A71 systemic infection is that EV-A71 first replicates in the gut and disseminates to the blood compartment or immune cells. Subsequent invasion of the CNS is mediated through infection of extraneural tissues, such as the muscle, to access motor neurons, and ascends to the CNS through retrograde transport, or by crossing the BBB to infect the brain (Figure 2). After the identification of SCARB2 as the main uncoating receptor for EV-A71, it remained to be understood how SCARB2 contributes to EV-A71 tropism and spread. To investigate this, several independent laboratories generated transgenic (Tg) mice expressing the human SCARB2 (hSCARB2) gene from a strong constitutive promoter [40,41,42]. Compared to wild-type mice, neonatal hSCARB2-Tg mice were more susceptible to subcutaneous infection by EV-A71 clinical isolates, and developed HFMD-like symptoms and neuropathogenesis. This corresponded with pro-inflammatory cytokine signaling and T lymphocyte infiltration, which is seen in severe EV-A71 disease [40]. Human SCARB2 expression greatly increased susceptibility to certain EV-A71 strains and allowed for infection of mice beyond the neonatal age and resulted in severe limb paralysis and death. In hSCARB2-Tg mice, EV-A71 viral particles were identified in the brainstem, spinal cord, intestine, skeletal muscle, and skin [40,41]. The increased susceptibility was observed upon infection of EV-A71 through multiple inoculation routes, suggesting that SCARB2 could function in diverse tissue types to promote pathogenesis in these mouse models. In humans, SCARB2 is widely expressed in different tissue types, suggesting that it may function as an important cellular factor broadly required for in vivo tissue tropism and neuropathogenesis.

In contrast to SCARB2, overexpression of human PSGL-1 in mice does not increase susceptibility to EV-A71 infection with clinical EV-A71 strains [43]. However, a mouse-adapted strain of EV-A71 with increased neurovirulence was shown to have enhanced usage of mouse PSGL-1 [44]. While PSGL-1 is primarily found on leukocytes, it has also been reported to be expressed in neurons and glial cells and may be important for EV-A71 infection of these tissues. [45]. This suggests that the ability to use PSGL-1 may contribute to the spread and replication of EV-A71 to the blood compartment and neuronal tissues and could play a pivotal role in tissue distribution that leads to neuroinvasion.

PHB may contribute to EV-A71 infection specifically in neuronal cells, as it was found to play a role in EV-A71 infection of human and mouse neuronal cell lines, but not in other cell types [26]. Rocaglamide, an anti-cancer drug that inhibits PHB, showed anti-viral properties in a mice model of EV-A71 infection. Rocaglamide specifically inhibited viral replication in the brain and spinal cord, whereas replication in the muscle was unaffected [26]. Vimentin is also implicated in EV-A71 neurovirulence. Vimentin is primarily expressed in endothelial cells and plays a critical role in cell adhesion and maintenance of cellular junctions. The EV-A71 capsid protein VP1 was found to increase the permeability of the BBB in both a mouse model of infection and an in vitro BBB model consisting of a monolayer of brain endothelial cells [46]. This increase in BBB permeability was suggested to be due to an effect of VP1 on increasing vimentin protein expression. A mutation in VP1 that decreases the interaction with vimentin also contributes to a reduced infection of CNS. [47] Vimentin likely plays a role in the dissemination of EV-A71 across the BBB from the blood compartment.

The extent to which heparan sulfate (HS) usage contributes to EV-A71 neurotropism in vivo is controversial. HS chains are linear polysaccharides that are covalently attached to proteins to form HS-proteoglycans and are expressed ubiquitously on most cell surfaces and in the extracellular matrix. A mutation in the VP1 capsid of an EV-A71 clinical isolate acquired from an immunocompromised host was shown to confer the ability to bind to HS [48]. This mutation contributed to high levels of replication in both intestinal and neural tissues, suggesting that binding to HS contributed to increased dissemination, allowing the virus to reach and infect the CNS. Additionally, the use of HS analogs has been shown to inhibit EV-A71 infection in vitro and may be a promising host targeted therapy [27], [49]. In stark contrast, several studies have shown that HS usage is correlated with an attenuation of infectivity in vivo. It was found that residue 145 in the VP1 region of EV-A71 was important for HS binding and dependence [50]. This residue is polymorphic in strains directly sequenced from patients, suggesting that HS utilization is a selectable trait [51]. The amino acid most frequently found (E) results in HS independence, while a minor variant (G) results in HS dependence. Utilizing isogenic strains with either residue at position 145, it was shown that weaker HS binding led to higher virulence and lethality in mouse models and in a Cynomolgus monkey model [52,53,54]. Enhanced HS usage may increase EV-A71 adsorption to peripheral tissues, thus preventing spread. Further studies are needed to determine the contribution of HS to dissemination of EV-A71 and tissue tropism. 

## 7. Contribution of Other Host Factors to Neurotropism and Pathogenesis

For enteroviruses that infect the GI tract, the microbiome has been suggested to play a role in viral dissemination and inter-host transmission. Mice depleted of their microbiome by antibiotic treatment were not able to support robust replication of either PV or CV-B3 in the intestine in a fecal–oral model of infection [55,56]. However, if antibiotic treated mice were infected IP with PV, bypassing replication in the gut, there was no difference in infection compared to mice harboring a normal microbiome [55]. This suggests that enteric EVs utilize the microbiome to promote infection at their initial site of replication in the intestine, which then promotes dissemination and spread to other tissues. The microbiome enhances PV infection in the gut due to virion binding to bacterial polysaccharides, supporting cell attachment, and increasing binding of the virion to its receptor [55,57]. The microbiome may also increase viral fitness by facilitating binding of multiple virions to one cell enabling viral co-infections [58]. Due to the fast error-prone nature of viral replication, EVs exist as a population of viruses with a variety of different genomes, including some mutations that may cause a fitness disadvantage. Initiation of infection by multiple genetically diverse virions can overcome deleterious mutations in the population and also allow for genetic recombination between different enteroviruses. In addition to promoting infection in the current host, the microbiome can enhance transmission between hosts by stabilizing EV virions. Binding of picornaviruses including PV and CV-B3 to bacteria or their cell surface polysaccharides was found to increase virion stability, which can increase chances of transmission to a new host by prolonging the viability of EVs in the environment [57,59]. The use of antibiotics may be a strategy to prevent dissemination and spread of enteric EVs. Antibiotics can reduce EV infection in the intestine after oral inoculation by reducing the number of bacteria in the gut. Additionally, antibiotics have been found to have a microbiota independent effect on viral infection. Antibiotic treatment in CV-B3 or PV infected mice injected IP or in germ free mice reduced viral titers and altered dissemination kinetics [60]. Antibiotics may reduce viral loads by modulating the host immune response or metabolic pathways. 

While pro-inflammatory cytokines induced in response to viral infections often help to restrict viral replication, they can also trigger the expression of pro-viral factors or result in host damage that increases disease severity. WARS is the first example of an IFN inducible entry factor for EV-A71 and upregulation during infection may promote virus entry into various tissues. IFN-γ production is stimulated following EV-A71 infection, resulting in increased expression of WARS and its translocation to the plasma membrane. WARS can sensitize cells that are lowly permissive to infection by EV-A71 such as neurons, and greatly increases the susceptibility to EV-A71 in combination with its primary receptor SCARB2. Interestingly, a hWARS transgenic mouse model of EV-A71 infection re-capitulated EV-A71 neuropathogenesis and skeletal muscle dissemination [25]. In humans, the severity of EV-A71 infection is correlated with an increase in proinflammatory cytokines including IFN-γ [61]. Since the IFN-γ receptor is ubiquitously expressed, WARS might contribute to widespread EV-A71 dissemination leading to severe neuropathogenesis. 

It is thought that brainstem damage, which leads to pulmonary edema and cardiopulmonary failure during severe EV-A71 infection, is caused by inflammation of the brainstem in response to virus infection rather than viral induced cytopathic effect. Severe EV-A71 infection is characterized by inflammatory damage to the CNS due to an activated cerebral immune response, with brainstem encephalitis and aseptic meningitis being the most common complications of severe infection. In support of this, EV-A71 infection was found to stimulate monoamine neurotransmitters and elevate cytokine levels in neonatal mice [62], and severe EV-A71 infection in human patients is associated with elevated levels of pro-inflammatory cytokines [61]. The inflammatory cytokine IL-6 plays a key role in the neuronal response to nerve injury and has been found to be involved in EV-A71 neuropathogenesis [63]. One possible way to prevent severe EV-A71 pathogenesis is through dampening the host immune response, preventing tissue damage as a result of inflammation. Inhibiting the inflammatory cytokine IL-6 with a neutralizing antibody increased the survival rate of EV-A71 infected mice [62]. Additionally, preventing brainstem dysfunction could be a viable approach to treating EV-A71 infections. EV-A71 infection was found to induce ER stress, leading to neuronal injury by exposing calreticulin on the surface of the cell, which marks the neurons for phagocytosis by microglia [64]. The use of glucocorticoids could alleviate ER stress and thereby reduce neuron damage [64]. Additionally, treating mice with phenoxybenzamine, an antagonist of the neurotransmitter adrenaline, increased their survival during EV-A71 infection [62].

## 8. Entry Factors

After internalization, to complete effective entry the virion containing endosome needs to properly traffic and mature to create an acidic environment thus inducing uncoating of the virus. Conformational changes in the capsid are induced by uncoating receptor binding and an acidic pH, leading to an expansion of the virion and exposure of VP4 and the N-terminus of VP1, which are initially located on the inside of the icosahedral capsid. The N-terminus of VP1 interacts with the endosomal membrane to anchor the virion, and VP4 forms membrane pores that allow for the translocation of the RNA genome into the cytosol. While EVs use a range of different receptors to initiate their internalization, there have been a few cellular factors identified that are needed by multiple EVs to reach the cytoplasm and complete the entry process.

Through a series of genome-scale genetic screens, the phospholipase PLA2G16 has emerged as a pan-enterovirus host factor required for infection of human cells and in in vivo mouse models of enterovirus pathogenesis. Although PLA2G16 was involved in virus entry, it was not required for the known steps including PV binding to the cell surface, endosomal trafficking, or pore formation [65]. Further genetic screening in a PLA2G16 deletion background revealed that knockout of distinct genes involved in a microbial clearance mechanism could rescue infection [66]. Endosomes permeated by VP4 are recognized by galectin-8 (LGALS8), targeting them for autophagic degradation, thus restricting virus infection. It was shown that both PLA2G16 and LGALS8 independently respond to membrane damage in uninfected and infected cells and are recruited to permeated endosomes that serve as sites of viral entry [65]. This suggests two opposing mechanisms upon viral entry, with PLA2G16 facilitating viral RNA release into the cytoplasm and inactivation of the virion in LGALS8-positive endosomes through autophagy. Despite its critical importance, the precise mechanism of how the phospholipase PLA2G16 facilitates RNA translocation is still incompletely understood. Since RNA release into the cytosol only occurs after a period of internalization, it is possible that PLA2G16 binding to permeated endosomes serves as a cue for the proper spatial and temporal release of the genome. 

A recent study identified three cellular factors involved in clathrin-mediated endocytosis and regulation of actin polymerization as promoting picornaviral entry: the kinase TNK2, its substrate WASL, and NCK1, which is an adaptor protein that functions to recruit WASL to TNK2 [67]. Knockout of these factors in several human cell lines reduced infection by encephalomyocarditis virus (EMCV). Other picornaviruses including CV-B3, PV and EV-D68 were also affected, albeit more modestly [67]. Mice deficient in TNK2 expression were partially protected against lethal EMCV infection and viral titers in the brain and the heart were reduced. Mechanistically, TNK2 was shown to affect virus internalization, while WASL acts downstream of endocytosis, as it did not affect virion binding to the cell, internalization of the virus, or pore formation [67]. Instead, knockout of WASL led to the accumulation of EMCV in early endosomes. It was proposed that TNK2 and WASL affect endosomal trafficking and maturation, which is necessary for subsequent release of the RNA genome.

## 9. IRES-Mediated Translation

Following RNA translocation into the cytosol, the (+) strand RNA genome is directly translated by the host machinery. Translation of the viral RNA yields a single polyprotein that is cleaved by the viral proteases 2A and 3C to generate 11 individual mature viral proteins and their precursors. Unlike translation of cellular mRNA, translation of viral RNA happens in a cap-independent manner, instead utilizing an internal ribosome entry site (IRES) in the 5′ UTR to initiate translation of the viral genome. The IRES is a highly structured region of the viral RNA and interacts with a number of cellular factors. IRES-mediated translation occurs through the combined action of a subset of canonical members of the host translation machinery, and so-called IRES trans-acting factors (ITAFs), which includes the RNA-binding proteins hnRNPA1, PTBP1, PCBP1/2, and FBP1, among others [68,69]. Many of these factors translocate from the nucleus to the cytoplasm during infection and aid in translation initiation by binding to the viral RNA and recruiting ribosomes. For comprehensive reviews of these ITAFs see references [68,69].

Novel ITAFs are continuing to be identified. For example, a DEAD-box family RNA helicase, DDX3X, was found to be important for IRES mediated translation of picornaviruses [70]. While the secondary structure of the IRES is thought to be important for IRES activity and recruiting cellular factors, the AUG that mediates ribosome entry is sequestered in a stem-loop structure, which decreases its accessibility to the ribosome and hinders viral translation. Thus, an additional protein is thought to be needed to unwind the secondary structure of the IRES RNA to facilitate ribosome recruitment. It is thought that DDX3X is recruited to the enterovirus IRES via an interaction with truncated EIF4G, and subsequently destabilizes the secondary structure facilitating ribosome entry and scanning. Recently, Staufen1 (STAU1) was found to be involved in EV-A71 translation. STAU1 is a double-stranded RNA binding protein that binds to cellular mRNAs to regulate mRNA translation and trafficking. STAU1 enhances mRNA translation by binding to their 5′ UTR and increasing the number of polysome containing mRNAs. STAU1 was found to bind to the 5′ UTR of EV-A71, and knockdown of STAU1 was found to inhibit IRES-mediated EV-A71 translation leading to decreased viral protein production and subsequent RNA replication [71]. STAU1 enhances viral translation efficiency by recruiting additional ribosomes onto the viral RNAs, resulting in the formation of polysome complexes and prolonging viral RNA stability [71].

The ITAFs promoting viral translation often have important roles for critical cellular functions in transcriptional regulation, splicing, RNA transport, RNA stability, or translational control. Thus, targeting these factors for anti-viral therapy may cause unintended side effects. 

## 10. Genome Amplification

Genome replication is a highly conserved process between different EVs, making it a suitable target for broad-spectrum HDT. The viral RNA-dependent RNA polymerase, 3D^pol^, makes up the core of the replication machinery and facilitates genome replication in combination with several host and viral proteins. Genome replication by 3D^pol^ starts with synthesis of a negative-strand (−) RNA that serves as a template for synthesis of new positive-stranded (+) RNA molecules that can be used for further rounds of translation, enter a new round of replication, or be packaged into progeny virions. The viral genome is bound by the viral protein VPg (3B), which is required as a primer for replication. Viral proteins 2C and 3CD aid in synthesis of (−) strand RNA by serving as a helicase and circularizing the RNA, respectively. Roles of host proteins in viral replication are often poorly understood, and elucidating the mechanisms of host proteins involved in viral replication should be a priority of future work.

Through unbiased knockout screening approaches, it was discovered that the protein methyltransferase SETD3 has an essential regulatory role for infection by a broad range of human EVs including different rhinoviruses, coxsackieviruses, poliovirus, and the neurotropic enteroviruses [72]. Human cells lacking SETD3 did not affect early steps of enterovirus infection such as entry, but severely hampered viral RNA genome replication. SETD3 is a methyltransferase that mono-methylates actin on the histidine 73 position, thereby regulating actin function [73,74]. However, methyltransferase activity of SETD3 was not required for its role in viral replication, indicating that SETD3 has a function in EV replication independent of actin methylation. SETD3 was found to bind to the viral nonstructural 2A protein of several enteroviruses. Mutations in 2A that prevented the interaction between SETD3 and 2A were lethal when introduced in the CV-B3 genome. SETD3 deficiency did not appreciably affect 2A protease activity on its viral substrate or the tested cellular substrates, suggesting that SETD3 promotes some unknown role of 2A in viral replication. SETD3 was critically important for in vivo pathogenesis, as it was shown that Setd3-/- neonatal mice were completely protected from lethal challenge with EV-A71 and CV-A10 upon intracranial injection. These findings demonstrate that SETD3 controls pathogenesis for a large class of enteroviruses with a strong impact on human health. SETD3 is a promising target for the development of HDT, as its role as a pan-enterovirus host factor would allow for the targeting of multiple different EV infections. Complete SETD3 deficiency in Setd3-/-mice did not lead to reduced viability of these animals, suggesting that short-term inhibition as needed for anti-viral therapy might have a good safety profile. 

Another host protein involved in viral RNA replication is UDP-glucose glycoprotein glucosyltransferase 1 (UGGT1). UGGT1 is normally localized to the ER, and regulates glycoprotein folding, playing a key role in the unfolded protein response (UPR). During EV-A71 infection, UGGT1 re-localizes to the viral replication compartments and interacts with the viral RNA polymerase 3D^pol^ [75] (Figure 3). A mutant of UGGT1 lacking monoglucosylation activity was as efficient in promoting EV-A71 infection as wild type UGGT1, suggesting a non-canonical role in promoting viral replication. Expression of the nonstructural protein 3A was shown to increase the amount of UGGT1 in membrane fractions. While the mechanism by which UGGT1 is pro-viral is not known, Huang and colleagues propose that UGGT1 might be recruited by the replication machinery to act as a bridge between the viral replication proteins 3A, 3AB, 3C, and 3D, as these proteins also co-purify with the UGGT1-3D complex. It is also possible that UGGT1 promotes the proper functioning of 3D^pol^. 

## 11. Replication Organelle Biogenesis and Phospholipid Biosynthesis

One of the most striking features of replication by positive-stranded RNA viruses is the extensive remodeling of host cell membranes to generate replication organelles (ROs) that serve as sites of viral genome replication. Enterovirus ROs are dynamic structures, initially appearing as single membrane vesicles early in infection and transitioning into double membrane vesicles and multilamellar vesicles later in infection, and eventually filling the majority of the cytoplasm [76,77]. ROs originate from endoplasmic reticulum (ER) and trans-Golgi membranes, leading to Golgi apparatus disintegration during enterovirus infection [78]. It has been proposed that ROs serve to concentrate both the RNA genome and host and viral proteins needed for efficient genome replication, mediate proper orientation of the replication machinery, and shield the replicating viral RNA from host immune responses [79]. 

In order to generate the massive web of ROs during infection, EVs induce the production of new lipids [80]. To synthesize new membrane lipids, EVs need to acquire long chain fatty acids (FAs) from the host, which serve as substrates in phospholipid biosynthesis. Emerging evidence suggests that the phospholipid synthesis that drives RO biogenesis during EV infection is primarily sustained by FAs released from lipid droplets (LDs) rather than directly utilizing imported FAs or FAs obtained from de novo synthesis [80,81]. LDs are important hubs of lipid metabolism that are generated in the ER and coordinate the synthesis, storage, and mobilization of neutral lipids. Upon PV infection, an increase in FA import from the extracellular media is observed that is dependent on the host protein acyl-CoA synthetase 3 (Ascl3) [82]. These imported FAs in turn get transferred into LDs, which are then utilized to support phospholipid biosynthesis of the developing ROs [80,81]. The viral 2A protein is necessary but not sufficient for stimulating FA uptake but, intriguingly, this is independent of its protease activity. Using electron microscopy, it was found that LDs form distinct membrane contact sites with the developing viral ROs [78,81,83]. FAs are released from LDs through the cleavage of triglycerides within the LD core by LD-associated lipases, including hormone-sensitive lipase (HSL) and adipose triglyceride lipase (ATGL) (Figure 3). It was shown that upon enterovirus infection, a cytoplasmic pool of host lipases is recruited to the surface of LDs, likely through interactions with several non-structural proteins (Figure 3) [80,81]. These lipases were found to play a key role in the release of FAs during EV infection, as their inhibition prevented the formation of ROs and subsequent RNA replication [80,81]. The released FAs stimulate the synthesis of structural phospholipids such as phosphatidylcholine (PC), which drive the massive expansion of the membranes. The rate-limiting enzyme in PC synthesis (CCTα) translocates from the nuclei of infected cells and associates with membranes of the viral replication complexes. This translocation requires the activity of the viral protease 2A. Altogether, these studies highlight that the ROs are not remnants of secretory organelles, but rather novel infection-specific organelles formed by co-opting cellular membrane pathways through specific virus–host interactions. Moreover, LD-associated kinases could be targets for antiviral therapy. HSL inhibition by the compound CAY10499 resulted in significantly reduced replication levels of rhinoviruses and poliovirus at concentrations that did not affect cell growth [80,81]. Inhibition of ATGL with atglistatin resulted in a modest decrease of poliovirus replication in cells [81]. 

## 12. Creating the Optimal Lipid Microenvironment for RNA Replication

It has become increasingly clear that the replication organelles induced during EV infection have a unique lipid composition [84]. In particular, phosphatidylinositol-4-phosphate (PI4P) and cholesterol were found to be critical for enabling RO development and RNA genome amplification. PI4P is synthesized in cells by phosphatidylinositol 4-kinases (PI4Ks). There are two types of PI4Ks, type II and type III, each with an A and B isoform that are differentially located at various cellular membranes where they produce PI4P locally. Enteroviruses replication is dependent on phosphatidylinositol 4-kinase IIIβ (PI4KB), a kinase that generates the primary pool of PI4P at Golgi membranes [85,86]. One proposed mechanism by which specific enterovirus non-structural proteins recruit PI4KB to ROs is by engaging GBF1. GBF1 is a guanine nucleotide exchange factor that activates Arf1 GTPase to recruit and activate PIK4B, thus co-opting components of the cellular secretory pathway [85,87,88]. Unbiased protein–protein interaction screens revealed a robust interaction between the Golgi adaptor protein ACBD3 with the 3A non-structural protein from several enteroviruses [89,90]. As an interaction partner of PI4KB, ACBD3 was shown to mediate recruitment of PI4KB to the ROs, and depletion of ACBD3 severely inhibited replication of multiple enteroviruses [89,90,91]. Recently, another PI4KB binding protein, c10orf76, was found to bind ACBD3 [92] and be required for replication of certain enteroviruses [93]. Knockout of c10orf76 resulted in decreased levels of PI4P at the Golgi and redistribution of GBF1 [86,93,94]. Thus, specific mechanisms have been identified by which enteroviruses recruit PI4KB to ROs to increase local PI4P levels by interactions of the 3A protein with PI4KB interaction partners. It is likely that other viral non-structural proteins also contribute to the increased PI4P levels during viral infection, since expression of either 2BC or 3CD poliovirus proteins were found to induce synthesis of PI4P [95,96]. 

Several roles for PI4P during viral replication have been proposed. The PV viral polymerase 3D binds specifically and preferentially to PI4P lipids in an in vitro assay [85]. In a separate study, a PI4P binding domain was mapped to the 3C protein, which overlapped with a known RNA binding site [97]. Thus, PI4P might act by directly interacting with viral proteins involved in RNA replication to facilitate interactions of the replication complex with membranes or viral RNA. Additionally, PI4KB was demonstrated to regulate the proper processing of the viral polyprotein at the junction between proteins 3A and 3B [98], suggesting a role for the proper lipid environment in cleavage of the viral polyprotein. Another role of PI4P is in driving a lipid flux that results in cholesterol recruitment to ROs. Cholesterol, presumably by allowing membrane curvature, is important for enterovirus replication, although this is not reliant on de novo biosynthesis [83,99]. Instead, cholesterol is mobilized from preexisting pools. This can be either by enhanced uptake from the plasma membrane by upregulating clathrin-mediated endocytosis [99,100], or through lipolysis from lipid droplets [81,83]. It was found that enteroviruses transfer cholesterol to the developing ROs by co-opting a cellular pathway in which cholesterol is transported from ER membrane to the Golgi membrane through a counterflux of PI4P (Figure 3) [38,95,101,102]. Central to the pathways is the lipid transfer protein, oxysterol-binding protein (OSBP), which forms membrane contact sites between the ER and trans-Golgi to exchange PI4P, and cholesterol between the two organelles. OSBP binds to ER membranes by interacting with ER localized VAP-A or VAP-B proteins and is anchored to the Golgi through PI4P [103]. During EV infection, OSBP is redirected to contact RO membranes where PI4P is highly abundant. OSBP transfers PI4P from ROs down its concentration gradient to the ER and in turn transports cholesterol to ROs. At the ER membrane, Sac1 hydrolyzes PI4P to phosphatidylinositol (PI), which can then be transferred back to the ROs by phosphatidylinositol transfer protein beta (PITPb). PI can then be used as a substrate to by PI4KB to produce PI4P.

The strong dependence of enterovirus replication on cellular pathways to build the replication organelles required for efficient replication provides a wealth of potential host targets suitable for host-directed therapy. Indeed, compounds targeting key enzymes in lipid metabolism have been shown to potently suppress replication of enteroviruses. A prominent target is the kinase PI4KB, which is required by a wide range of enteroviruses. Several inhibitors targeting this enzyme have broad-spectrum activity in human cells and in mouse models of enterovirus infection [38,85,104,105,106,107]. Likewise, OSBP, the protein important for PI4P/cholesterol flux, was shown to be inhibited by several potent antiviral compounds, further highlighting this pathway as a potential host-directed target [101,108,109,110,111,112]. Enteroviruses can be selected with mutations that allow replication in PI4K knockout cells and in wild type cells treated with PI4K/OSBP targeting compounds [106,113,114]. Most mutations map to the 3A protein, further reaffirming the critical role of this protein in recruiting PI4KB. Development of PI4KB inhibitors for antiviral drug therapy has been hindered by an antiproliferative effect in lymphocytes and in vivo mortality in mice at the tested concentrations [115,116]. The proposed OSBP inhibitor, itraconazole, displayed no reported toxicity upon intranasal administration and in a mouse model of rhinovirus infection there was reduced viral replication in lungs [117]. 

## 13. Autophagy in the EV Life Cycle

EVs subvert and exploit the cellular autophagy pathway during distinct steps of the viral life cycle to facilitate efficient infection. Autophagy is an intracellular degradation pathway that recycles cellular components to maintain cellular homeostasis and is controlled by a set of autophagy-related (ATG) genes. Traditionally, autophagy goes through a series of specific steps, starting with nucleation of an initial double membrane structure through the action of several protein complexes, and subsequent membrane elongation and maturation to form a fully enclosed double membrane organelle or autophagosome. Autophagosomes enclosing cytoplasmic contents fuse with lysosomes, leading to the degradation of their cargo for reuse by the cell. Autophagy can also facilitate selective clearance of materials in response to cellular and environmental stimuli such as invading pathogens. Upon virion internalization, EV RNA is translocated through a pore in the endosome to reach the cytoplasm where it can be translated and replicated. Pore formation triggers the recruitment of LGALS8, which targets the permeated endosomes for autophagic degradation. It was proposed that the phospholipase PLA2G16 facilitates the displacement of the viral genome from permeated endosomes, thus allowing the RNA to reach the cytoplasm before autophagy-mediated degradation of the viral RNA can occur [65]. In contrast to entry, during genome replication, EVs intentionally trigger autophagy to upregulate the formation of double membrane autophagosomes that provide additional membrane surfaces for intracellular replication [118,119]. PV employs a noncanonical autophagy pathway, since it utilizes some of the autophagy initiation components, but does not require the entire pathway [120]. PV requires a subset of autophagy proteins for its infection including ULK1 and FIP200, which act in a complex together to regulate the initiation of autophagosome formation, ATG9, which scavenges lipids for the elongating membrane, and LC3, which recruits cargo to autophagosomes, regulates membrane curvature, and is necessary for the closure of the autophagosome [120]. Additionally, PV infection can lead to the activation of AMPK, which activates ULK1 and the downstream autophagy pathway [120]. Canonical autophagy requires ATG5 to recruit LC3 to the autophagosome; however, during PV infection, ATG5 is dispensable, suggesting some other mode of recruiting LC3 directly to membranes [120].

The EV life cycle ends with the encapsidation of nascent viral RNA and assembly of mature virus progeny that are capable of infecting new cells. Several recent studies have shown that EVs are released from cells in vesicles containing pools of virions through a non-lytic pathway. It is thought that EVs upregulate autophagy to engulf clusters of virions to aid in en bloc transmission of new virions prior to cell lysis. During EV infection, multiple infectious viral particles have been shown to be captured within double-membrane organelles that resemble autophagosomes. These vesicles are enriched in phosphatidylserine (PS) [121,122], contain the autophagy marker LC3 [118,119,121], and are thought to originate from sections of ROs, which were also found to be enriched in PS [121]. The presence of LC3 on these vesicles may aide in the sequestration of newly formed virions as EV capsid proteins have been shown to bind to LC3 [120]. In order to be released from the cell, EVs must prevent the fusion of these autophagosome-like vesicles with lysosomes to avoid the degradation of newly formed virions. It is thought that this is achieved by disrupting the interaction between SNARE complex proteins SNAP29, STX17, and VAMP8 that mediate the fusion between autophagosomes and lysosomes [118,121,123]. Instead, these double membrane vesicles fuse with the plasma membrane secreting single membrane vesicles containing infectious virions. The presence of PS on these vesicles increases their uptake by neighboring cells, allowing for spread of the multiple virions into a new host cell. Infection of new cells by these secreted vesicles is still dependent on the individual EV proteinaceous uncoating receptor, in addition to the use of PS as a co-factor [121]. Secreted infectious vesicles were demonstrated to be more efficient at infecting new cells than free EV particles, possibly by delivering multiple viral genomes to a single cell to increase the genetic diversity and multiplicity of infection, or by cloaking virions from neutralizing antibodies. A recent study found that infectious vesicles secreted during PV infection can also transport viral un-encapsidated RNAs (both positive and negative strand RNA), viral non-structural proteins, and host proteins, in addition to mature virions [122]. This may accelerate viral replication in newly infected cells by delivering components needed for replication, prior to their generation during initial translation. In addition to the use of autophagosomes, EVs may also utilize components of exosomes to facilitate en bloc transmission of virions to new cells both in vitro and in vivo [119,122,124,125]. Interestingly, non-lytic cell release may be a major mechanism of spread in the CNS as EV-A71 infection of neuronal cells (derived from human neural stem cells), neuroblastoma IMR-32 cells, or NSC-34 cells induces autophagy, yet shows no apoptosis or cytopathic effect [119,126]. Inhibition of autophagy by 3-MA led to a decrease in viral load in the brain after intracranial injection of EV-A71 in suckling ICR mice, suggesting that targeting autophagy may be a viable option to prevent neurodegeneration [127]. 

## 14. Conclusions and Future Perspectives

Enteroviruses are among the most common causes of infection in humans but typically are asymptomatic or cause mild illnesses like the common cold. Serious complications can occur in rare instances, especially in infants and people with weakened immune systems. Importantly, in recent years, enterovirus strains EV-D68 and EV-A71 have emerged with increased neurovirulence, likely causing outbreaks of acute flaccid myelitis, a severe polio-like paralytic disease that affects children [12]. This highlights that enteroviruses can rapidly acquire traits, changing from viruses causing relatively mild disease into viruses that pose a severe threat to human health. Identifying common cellular pathways required by multiple EVs could expose vulnerabilities amenable for host-directed antiviral therapy against a broad spectrum of EVs, including newly arising strains. Moreover, identifying host proteins specific to the neurovirulent strains EV-D68 and EV-A71 may allow us to better understand factors contributing to the neurotropism of these viruses.

The identification of cellular receptors for EV-D68 and EV-A71 provided valuable insights in the mechanisms of viral entry of these significant pathogens. Sialic acid and ICAM-5 emerged as the main receptors for EV-D68, although sialic acid independent strains have been isolated from clinical samples and ICAM-5 has a restricted expression pattern and is absent in cell types critical for viral spread such as the distal axons of motor neurons. Important questions remain on the role of cellular entry factors in EV-D68 pathogenesis. Is there a yet unknown receptor used by contemporary strains to access the CNS? What underlies the apparent increased neurovirulence of current EV-D68 strains? For EV-A71, the main functional entry receptors are SCARB2 and PSGL1. SCARB2 is widely expressed, promotes uncoating of the virion, and expression of human SCARB2 in mice dramatically enhances their susceptibility to EV-A71 infection, suggesting it plays an essential role in EV-A71 infection, similar to the role of the poliovirus receptor (PVR) in PV infection. PSGL1 is expressed preferentially on leukocytes and is likely important for EV-A71 organismal spread. More research is needed to shed further light on the role of these and the other entry factors in facilitating neuroinvasion and neuropathogenesis. For both neurotropic viruses, an intriguing question is to what extent do immune responses contribute to viral spread and damage to neural tissues? Finally, it has been found that enteroviruses directly interact with the microbiota in the gut to promote viral replication. Future research into how microbiota–immune interactions might determine enterovirus pathogenesis will provide another trans-kingdom viewpoint on virus–host interactions [128]. 

For host-directed antiviral therapy, cellular targets that are essential for infection of a wide range of enteroviruses could lead to development of broad-spectrum drugs. Recent studies focusing on the cellular factors involved in lipid acquisition and replication organelle biogenesis have elucidated promising candidates. Indeed, compounds targeting OSBP and PI4KB have shown to be antiviral for multiple enteroviruses, although for PI4KB inhibition, cellular toxicity and mortality in mice was reported. Genome-scale knockout screens have identified additional cellular targets that are essential for viral entry and replication, while not affecting cellular or organismal viability. For example, knockout mice deficient for *setd3*, *pla2g16*, and *tnk2* were viable and were strongly protected against lethal challenge with different picornaviruses. Development of compounds targeting these cellular factors has the potential to yield potent antiviral drugs with minimal toxicity expected from short-term inhibition. The unbiased discovery of these cellular factors has not only revealed novel molecular mechanisms by which enteroviruses co-opt the cellular machinery to promote replication and spread, but also ways in which we might, in the future, combat severe enterovirus disease. 

## Figures and Tables

**Figure 1 viruses-13-00166-f001:**
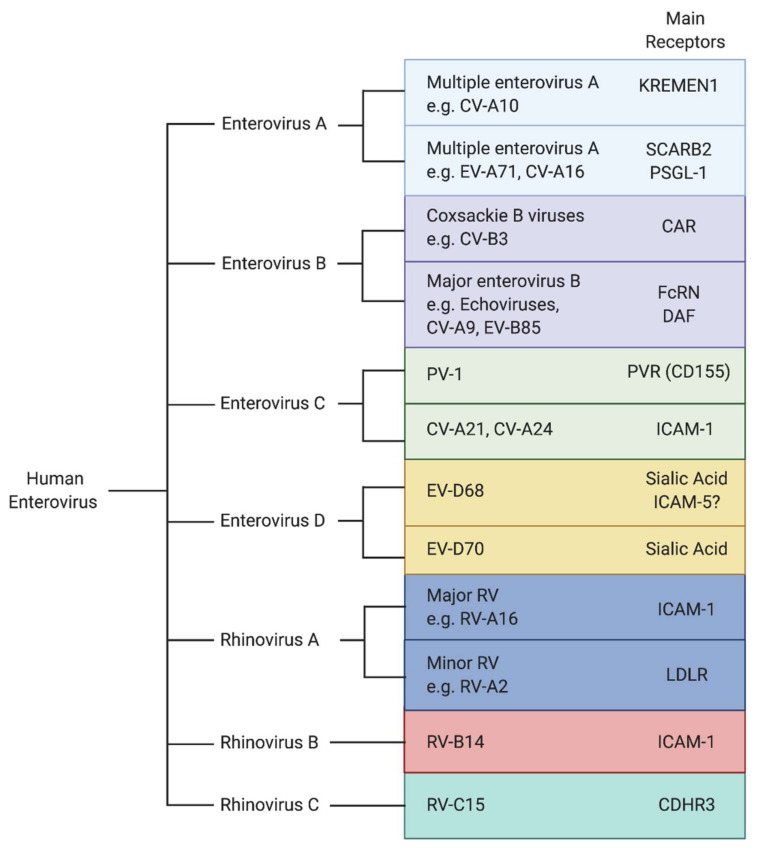
Main receptors for human enteroviruses. Enteroviruses A–D contain neurotropic viruses that are associated with encephalitis, meningitis, and acute flaccid paralysis. Rhinoviruses A–C contain respiratory viruses that are the causative agents of the common cold. Representative EV serotypes are named here.

**Figure 2 viruses-13-00166-f002:**
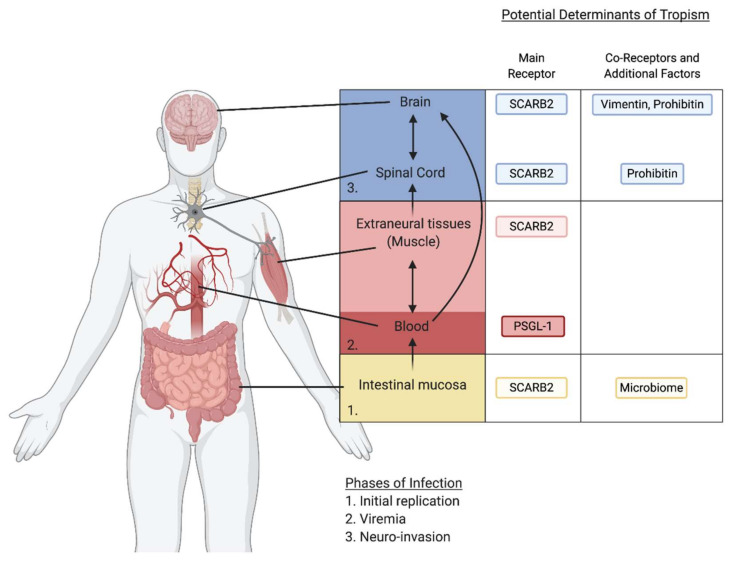
Speculative model of EV-A71 spread from the primary site of replication in the intestine to the CNS. There are three proposed phases of EV-A71 systemic infection: (1) initial replication in the gut, (2) dissemination to the blood compartment or immune cells, and (3) subsequent invasion of the CNS by infecting extraneural tissues to access motor neurons or by directly crossing the BBB. EV-A71 utilizes many receptors to promote its infection of various tissues. SCARB2 and PSGL-1 are the primary receptors. SCARB2 is broadly required for dissemination to many tissues, while PSGL-1 is found primarily on leukocytes and is important for the viremia phase of infection. Additional receptors including Vimentin and Prohibitin have been implicated in EV-A71 infection of the CNS, while the microbiome plays a role in initial infection of the gut leading to widespread EV-A71 infection.

**Figure 3 viruses-13-00166-f003:**
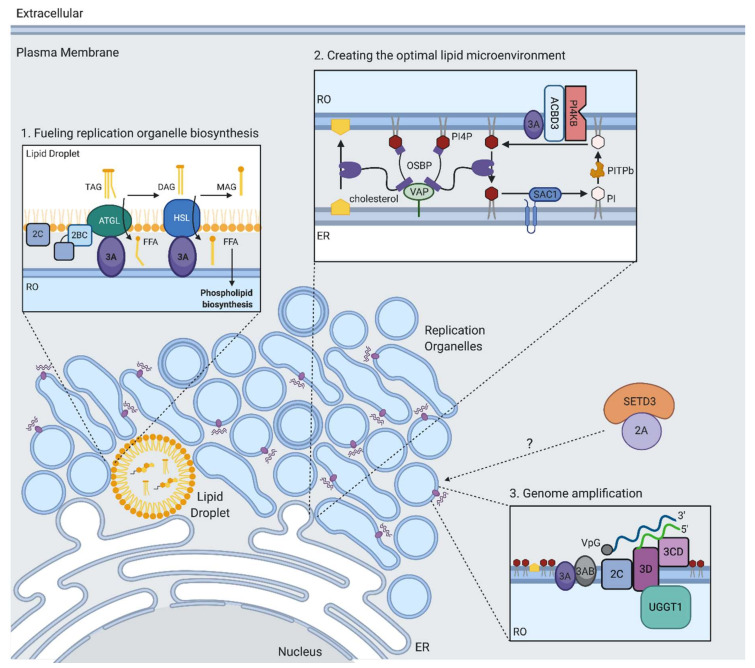
Cellular factors required for viral replication. (1) Genome replication takes place on virus induced membranous replication organelles (ROs). To establish the massive web of replication organelles during infection, EVs upregulate phospholipid biosynthesis. Upon infection, the import of free fatty acids (FFAs) and their storage in lipid droplets (LDs) is increased. FFAs serve as substrates for phospholipid biosynthesis. EVs scavenge FFAs from lipid droplets to sustain the production of new lipids that drive RO biogenesis. (2) EV viral proteins recruit host proteins to generate a favorable lipid environment on developing ROs for viral replication. Viral proteins recruit PI4KB to enrich replication organelle membranes with the phospholipid PI4P. OSBP is redirected to ROs to transport cholesterol from the ER by shuttling PI4P down its concentration gradient. At the ER membrane, Sac1 hydrolyzes PI4P to phosphatidylinositol (PI), which is transferred back to the ROs by PITPb. (3) The viral RNA polymerase 3D^pol^ replicates the viral genome in coordination with several host and viral proteins. The viral genome is bound by the viral protein VPg (3B), which is required as a primer for replication. Viral proteins 2C and 3CD aid in replication by serving as a helicase and circularizing the RNA. Genome replication by 3D^pol^ begins with the synthesis of a negative-strand (−) RNA that serves as a template for the synthesis of new positive-strand (+) RNA. The host factors SETD3 and UGGT1 interact with viral proteins to promote RNA replication; however, their exact mechanisms are unknown.

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
