# Peer review of "Return of the Neurotropic Enteroviruses: Co-Opting Cellular Pathways for Infection"

_viruses, 2021, doi:10.3390/v13020166_

Round 1
Reviewer 1 Report
Enteroviruses are emerging viruses of concern. This review article places the fundamentals of the enterovirus lifecycle in the context of conserved mechanisms and host pathways used by the entire genus. The overarching theme is that host-directed therapeutics with broad-spectrum activity may be feasible for these viruses.
The manuscript is well written. The illustrations are clear and effective. The manuscript can be accepted as is.
Author Response
Point 1: The manuscript is well written. The illustrations are clear and effective. The manuscript can be accepted as is.
Response 1: We thank the reviewer for their positive response.
Reviewer 2 Report
The authors summarized detailed the host cellular receptors involved with enterovirus entry and uncoating process and explained that the relationship of receptors and viral neurotropism. The host factors involved in IRES-mediated translation, viral RNA replication and replication organelle formation are all summarized detailed in this review paper. The readers can get much information from this review paper.
The suggestions are as follows:
- ITAFs of enterovirus have been studied very well. However, the information about the ITAFs of enterovirus showing in this review is not complete. The authors may use a table to summarize the identified ITAFs of enterovirus and their functions in regulation of translation of enterovirus.
- In “Autophagy in the EV life cycle” section, the authors may discuss the effect of autophagy in the enterovirus-infected neural cells.
- Lane 54, there are at least two vaccines against polio, Sabin vaccine and Salk vaccine.
4. Lane 185, 389, 396, please explain what is IFN-y?
Author Response
Point 1: ITAFs of enterovirus have been studied very well. However, the information about the ITAFs of enterovirus showing in this review is not complete. The authors may use a table to summarize the identified ITAFs of enterovirus and their functions in regulation of translation of enterovirus.
Response 1: We thank the reviewer for their comment. We agree that ITAFs are cellular factors with well-described functions in the enterovirus life cycle. As such, they have been the topic of comprehensive prior reviews. Because we would like to focus our review on more recently identified ITAFs we kept the discussion on more classic ITAFs short. In our revision we have now added a sentence to explicitly direct the reviewer to comprehensive reviews of ITAFs. See line 468: “For a comprehensive review of these ITAFs see ref68,69.” We have focused on the more recently identified ITAF Staufen 1 and in our revision we are now also including a discussion on the ITAF DDX3X, which has recently been discovered. See line 470: “Line 470: For example, a DEAD-box family RNA helicase, DDX3X, was found to be important for IRES mediated translation of picornaviruses [70]. While the secondary structure of the IRES is thought to be important for IRES activity and recruiting cellular factors, the AUG that mediates ribosome entry is sequestered in a stem-loop structure, which decreases its accessibility to the ribosome and hinders viral translation. Thus, an additional protein is thought to be needed to unwind the secondary structure of the IRES RNA to facilitate ribosome recruitment. It is thought that DDX3X is recruited to the enterovirus IRES via an interaction with truncated EIF4G, and subsequently destabilizes the secondary structure facilitating ribosome entry and scanning.”
Point 2: In “Autophagy in the EV life cycle” section, the authors may discuss the effect of autophagy in the enterovirus-infected neural cells.
Response 2: We appreciate this suggestion and have added discussion of relevant literature. In line 705 we have added: “Interestingly, non-lytic cell release may be a major mechanism of spread in the CNS as EV-A71 infection of neuronal cells (derived from human neural stem cells), neuroblastoma IMR-32 cells, or NSC-34 cells induces autophagy yet shows no apoptosis or cytopathic effect [119], [126]. Inhibition of autophagy by 3-MA led to a decrease in viral load in the brain after intracranial injection of EV-A71 in suckling ICR mice, suggesting that targeting autophagy may be a viable option to prevent neurodegeneration [127].
Point 3: Lane 54, there are at least two vaccines against polio, Sabin vaccine and Salk vaccine.
Response 3: We thank the reviewer for pointing out this oversight. In line 54 we have changed it to say: “With the exception of vaccines against PV and two vaccines against EV-A71 recently brought to market in China, there are no approved antivirals or therapeutics to treat or prevent enterovirus infections.”
Point 4: Lane 185, 389, 396, please explain what is IFN-y?
Response 4: In lane 186 we have changed IFN-y to the Greek symbol for gamma. In lane 185-186 we have written out: “WARS was found to be an interferon (IFN) inducible entry factor for EV-A71 with interferon gamma (IFN-g), a type II interferon, upregulating cell surface expression of WARS.”